# Voltage-dependent gating of SV channel TPC1 confers vacuole excitability

Dawid Jaślan[1], Ingo Dreyer [2], Jinping Lu[1], Ronan O'Malley[3,5], Julian Dindas[1,6], Irene Marten[1] & Rainer Hedrich [1,4]

In contrast to the plasma membrane, the vacuole membrane has not yet been associated with electrical excitation of plants. Here, we show that mesophyll vacuoles from Arabidopsis sense and control the membrane potential essentially via the $K^+$-permeable TPC1 and TPK channels. Electrical stimuli elicit transient depolarization of the vacuole membrane that can last for seconds. Electrical excitability is suppressed by increased vacuolar $Ca^{2+}$ levels. In comparison to wild type, vacuoles from the *fou2* mutant, harboring TPC1 channels insensitive to luminal $Ca^{2+}$, can be excited fully by even weak electrical stimuli. The *TPC1*-loss-of-function mutant *tpc1-2* does not respond to electrical stimulation at all, and the loss of TPK1/TPK3-mediated $K^+$ transport affects the duration of TPC1-dependent membrane depolarization. In combination with mathematical modeling, these results show that the vacuolar $K^+$-conducting TPC1 and TPK1/TPK3 channels act in concert to provide for $Ca^{2+}$- and voltage-induced electrical excitability to the central organelle of plant cells.

[1] Institute for Molecular Plant Physiology and Biophysics, University of Würzburg, Julius-von-Sachs Platz 2, 97082 Würzburg, Germany. [2] Centro de Bioinformática y Simulación Molecular (CBSM), Universidad de Talca, Talca 3460000, Chile. [3] Salk Institute for Biological Studies, La Jolla, CA 92037, USA. [4] Zoology Department, College of Science, King Saud University, PO Box 2455Riyadh 11451, Saudi Arabia. [5] Present address: DOE Joint Genome Institute, Walnut Creek, CA 94598, USA. [6] Present address: Department of Plant and Microbial Biology, University of Zürich, 8008 Zürich, Switzerland. Correspondence and requests for materials should be addressed to I.D. (email: idreyer@utalca.cl) or to R.H. (email: hedrich@botanik.uni-wuerzburg.de)

Recordings of transient changes in electrical potential with numerous plants have provided ample evidence that the plasma membrane of land plants is excitable, a feature usually only attributed to specialized cells in the animal nervous system[1]. In animals, the action potential is carried by voltage-dependent cation channels[2]. So far, there is little information available regarding the ion channels underlying plant excitability, and knowledge about the role of the vacuole in electrical signaling is even more scant[1]. Excitability of the vacuole membrane was, however, recognized in Charophyceaen freshwater green algae. In Chara, the tonoplast action potential is coupled to the excitation of the plasma membrane via a cytosolic $Ca^{2+}$ signal[3,4], which stimulates the release of $Cl^-$ ions from the vacuole and in turn leads to a transient vacuole polarization. The molecular nature of the ion channel activities underlying the Chara-type action potentials at the plasma membrane and vacuole membrane is hitherto unknown.

In the 1980s, patch clamp studies with higher plants identified a $Ca^{2+}$-dependent Slowly activating Vacuole channel that was named the SV channel[5,6]. Two decades later, in *Arabidopsis thaliana*, the single copy gene *TPC1* was shown to encode the SV channel[7], a finding confirmed by studies in rice[8]. This $Ca^{2+}$- and depolarization-activated non-selective cation channel can be found in the genomes of all vascular plants examined so far, and in moss, too[9]. TPC1-type channels are even found in animals; they are active, for instance, in the endosomes of human cells[10]. Plant and animal TPCs share sequence similarity to voltage-gated $Ca^{2+}$ and $Na^+$ channels and harbor two homologous Shaker-like domains (DI, DII) in tandem, each consisting of six transmembrane segments (S1-6 in DI and S7–12 in DII) and a pore-forming domain (P1, P2)[11,12]. The activity of plant TPCs is modulated by voltage and rise in cytosolic $Ca^{2+}$ level; they are thus potential entities for $Ca^{2+}$- and voltage-mediated signaling. For channel activation in response to rising cytoplasmic $Ca^{2+}$ levels, plant (but not animal) TPCs contain two $Ca^{2+}$-binding EF-hands in the linker region that connects the two Shaker-related modules[7,11–13]. In addition to TPC1 channels, tandem-pore potassium (TPK) channels are also equipped with EF hand motifs. They are activated upon a rise in the cytosolic $Ca^{2+}$ concentration[14,15], and in part contribute to the vacuolar $Ca^{2+}$-dependent $K^+$ conductance[15,16]. Among the four vacuolar-localized TPK family members, TPK1 and TPK3 are expressed in mesophyll cells (Arabidopsis eFP browser, http://bar.utoronto.ca/efp/cgi-bin/efpWeb.cgi).

Here, we document the electrical excitability of the vacuole membrane in higher plants. Electrical stimulation of Arabidopsis mesophyll vacuoles requires the presence of potassium ions, as well as expression of the $K^+$ channels TPC1 and TPK1/TPK3.

## Results

**Excitability of the vacuole membrane**. The polarization of the vacuole membrane was studied in the whole-vacuole configuration of the patch clamp technique, by employing the current clamp mode under $K^+$-based symmetrical solute conditions in the vacuolar lumen and cytosol (pipette and bath solution, respectively). Under these conditions the membrane voltage rests at zero mV, the Nernst potential for $K^+$ ($E_K$)[17]. However, *in planta*, under non-stimulated conditions, the plant vacuole exhibits a hyperpolarized resting membrane voltage[18], accomplished by two proton pumps. The V-type ATPase and $PP_i$ase actively translocate $H^+$ into the vacuolar lumen powered by the consumption of ATP and $PP_i$, respectively[19]. Therefore, to mimic the proton-pump-adjusted hyperpolarized ground state, currents were injected to maintain a negative resting voltage (−60 mV). To subsequently excite the vacuole membrane, a 200-ms-lasting depolarizing current pulse of +70 pA was injected. In response, the membrane voltage of wild type vacuoles was polarized in a process exhibiting three characteristic phases (Fig. 1a). Initially, the injected currents rapidly depolarized the membrane, appearing as a voltage spike with peak amplitude of approximately +65 mV (Fig. 1a, b). This spike phase was followed by relaxation to a depolarized plateau of about +14 mV, lasting until stimulus offset. Upon returning to pre-stimulation settings, the vacuole voltage did not immediately approach the −60 mV level but remained close to 0 mV (Nernst potential $E_K$) for about half a second, before the potential relaxed to the initial −60 mV state again. When the stimulus strength was varied from 10 pA up to 1000 pA, the amplitude of the initial voltage spike and the stimulus plateau voltage increased (Fig. 1a, b). Additionally, the duration of the post-stimulation depolarization plateau prolonged in a stimulus-dependent manner (Fig. 1a, c). The stimulation interval from 10 to 1000 pA is well within the physiologically relevant range. At the lower end it is in the range of vacuole ATP-driven $H^+$ pumps, which can generate up to 60 pA[20], while at the upper end it is about 5- to 10-fold smaller than the usual TPC1 currents measured under voltage clamp conditions[21].

**TPC1 is a prerequisite for vacuole membrane excitability**. Two types of $Ca^{2+}$-activated $K^+$-permeable channels are targeted to the plant vacuole membrane: TPC1/SV and TPK channels[16,22]. To identify the molecular entity giving rise to vacuole excitability, we initially inspected the TPC1-loss-of-function mutant *tpc1-2*[7]. Using *tpc1-2* mesophyll vacuoles, current injections from 10 up to 1000 pA caused voltage spikes similar to those monitored with wild-type vacuoles. Electrical stimuli depolarized the vacuole membranes of wild-type and *tpc1-2* mutant to a similar degree. This is related to the fact that at the resting membrane voltage TPC1 channels were initially closed (Supplementary Fig. 1a). During the stimulation episode, however, in wild-type vacuoles TPC1 channels activated after a delay, causing a partial repolarization. In contrast, vacuoles lacking TPC1 activity remained depolarized at the initial level throughout the entire stimulation phase. Thus, electrical stimulation of *tpc1-2* vacuoles elicited neither relaxation of the voltage spike to a stimulus plateau nor a post-stimulus depolarization (Fig. 2a, d and Supplementary Fig. 1a, b).

The electrical phenotype of *tpc1-2* vacuoles points to the TPC1/SV channel as a key factor for reaching the post-spike stimulus plateau and the following post-stimulus plateau. To further confirm this notion by an independent approach, in the wild type background the voltage-dependent activation of the TPC1/SV channel was inhibited by applying either $Ca^{2+}$ to the vacuolar lumen or by replacing $K^+$ with $Na^+$ on both sides of the vacuole membrane. Both cations shifted the voltage threshold for TPC1 channel activation to more positive potentials[23,24], leading to a decrease in steady-state current densities compared to $K^+$ (e.g., at +100 mV by 71% in 1 mM $Ca^{2+}$ and 87% in $Na^+$; Supplementary Fig. 1c, d). The inhibitory effect of luminal $Ca^{2+}$ on voltage-dependent TPC1 channel gating is linked to a $Ca^{2+}$ binding site at the luminal side of the TPC1 channel protein[11,12,23]. Therefore, when the luminal $Ca^{2+}$ concentration was increased to 1 mM, steady-state TPC1/SV currents significantly dropped (Supplementary Fig. 1c, d). Excitation experiments under these high luminal $Ca^{2+}$ conditions caused similar voltage responses to low current injections with wild-type vacuoles as observed with vacuoles from the TPC1-loss-of-function mutant *tpc1-2* in the absence of luminal $Ca^{2+}$ (Fig. 2a, b, d, e and Supplementary Fig. 1a, b). The stimulus spike and stimulus plateau were comparable in amplitude. A pronounced post-stimulus depolarization plateau was not elicited. However,

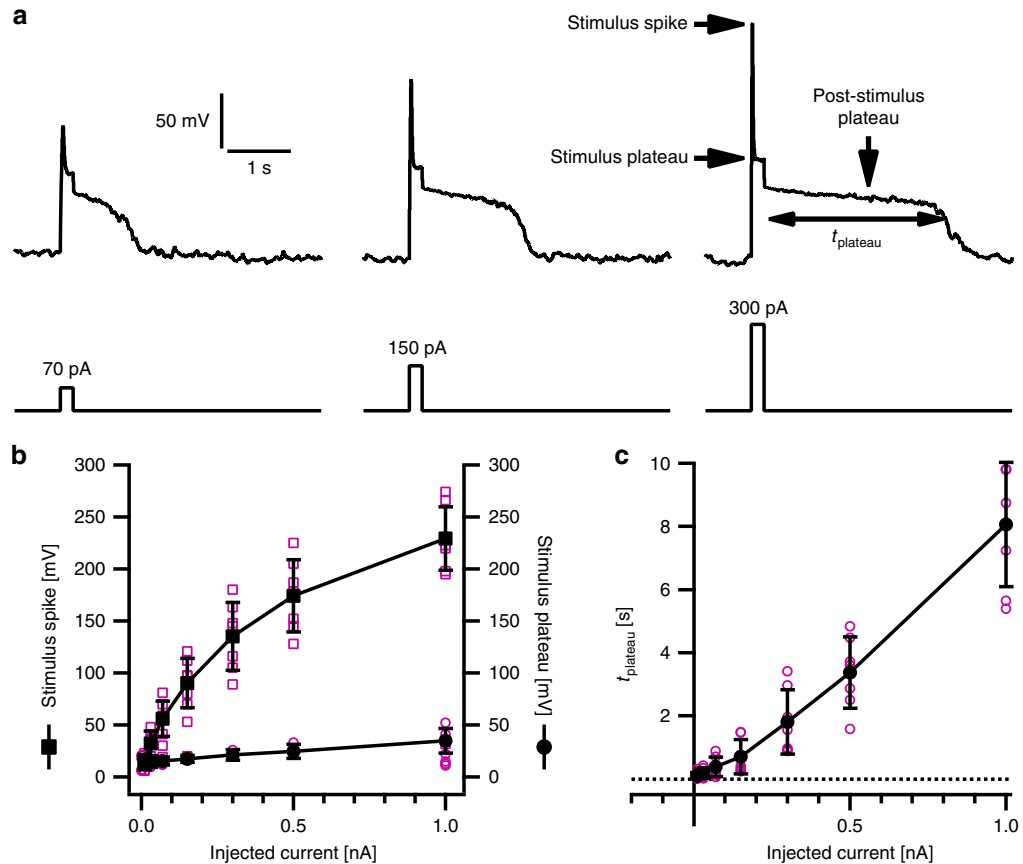

**Fig. 1** Electrical excitability of *Arabidopsis thaliana* wild-type mesophyll vacuoles. **a** Representative membrane voltage responses (upper panel) of vacuoles to current injections of 200 ms duration (70, 150, 300 pA, lower panel). As indicated at the top right, the current-induced voltage responses were analyzed with respect to the stimulus spike, the stimulus plateau and the lifetime of the post-stimulus plateau phase ($t_{plateau}$). **b**, **c** Amplitudes of triggered stimulus spike (**b**, left axis, squares), stimulus plateau (**b**, right axis, circles) and lifetime of the post-stimulus depolarized plateau phase (**c**) plotted against the corresponding injected current. Current clamp experiments in **a**–**c** were carried out with individual vacuoles under K⁺-based standard solutions. Closed symbols and error bars in **b**, **c** represent means ± standard deviation. Individual data points in **b**, **c** are given by open magenta symbols. Number of experiments in **b**, **c** was $n = 7$. Source data are provided as a Source Data file

with increased stimulus strength (>150 pA) the stimulus plateau amplitudes triggered from vacuoles exposed to luminal high Ca²⁺ were smaller compared to *tpc1-2* and became closer to those recorded from wild type vacuoles in the absence of luminal Ca²⁺ (Supplementary Fig. 1b). Thus, stronger current injections appeared to cause an increase in membrane conductance, but this rise was obviously still not enough to trigger pronounced post-stimulus depolarization plateaus, even not with the highest level of current injection (1000 pA) (Fig. 2e). Instead, only short-lasting post-stimulus depolarization phases were detected with wild type in the presence of 1 mM luminal Ca²⁺ (e.g., at 1000-pA injection: $t_{plateau\ WT\text{-}0Ca} = 8.06 \pm 1.96$ s, number of experiments $n = 7$, $t_{plateau\ WT\text{-}1Ca} = 0.14 \pm 0.16$ s, $n = 5$, means ± standard deviation; Fig. 2e).

Similar to elevated luminal Ca²⁺, Na⁺ loading in the vacuole also prevents voltage-dependent TPC1/SV channel activation[24]. Accordingly, Na⁺ shifted the activation threshold to more positive voltages (Supplementary Fig. 1c, d); and, as expected for impaired TPC1/SV channel function, the presence of luminal and cytosolic 150 mM Na⁺ had similar effects on excitability as shown for 1 mM luminal Ca²⁺. Current injections of 10 pA triggered comparable stimulus spike and stimulus plateau amplitudes in Na⁺- and K⁺-exposed vacuoles (Supplementary Fig. 1g, h). Upon stronger current injections, a relaxation from the stimulus spike to a depolarized stimulus plateau became apparent also in Na⁺-based solutions, but was less pronounced

than in the presence of K⁺. While a 300-pA stimulus induced similar stimulus spike amplitudes (i.e., +135 mV and +150 mV in K⁺ and Na⁺ solution, respectively), the voltage subsequently decayed to a stimulus plateau amplitude of +21 mV in K⁺ and to +84 mV in Na⁺ (Supplementary Fig. 1g, h). Thus, the stimulus-induced rise in the membrane conductance was lower in the presence of Na⁺ than in K⁺. In other words, with Na⁺ the stimulus was not strong enough to maintain the vacuole membrane in a depolarized state after stimulus offset (Supplementary Fig. 1f). Even current injections of up to 1000 pA did not trigger a post-stimulus depolarization, a behavior which we also observed with the *tpc1-2* mutant (Fig. 2a, d and Supplementary Fig. 1e, f). The absence of a pronounced post-stimulus depolarization phase in the presence of symmetrical Na⁺ or luminal Ca²⁺ suggests that the low number of TPC1/SV channels activated during stimulation were incapable of triggering a sustained post-stimulus depolarization.

After having studied the effect of shifting the voltage-dependent TPC1/SV channel activation towards more positive voltages, we addressed the question of how membrane excitability is affected when instead the TPC1 voltage dependence is shifted towards more hyperpolarized potentials. The *fou2* point mutation, D454N in TPC1, impairs the luminal Ca²⁺ coordination site[17,23,25]. As a result, the TPC1/SV channel becomes not only insensitive towards inhibitory luminal Ca²⁺, but shows a fundamental change in its voltage-dependent gating features.

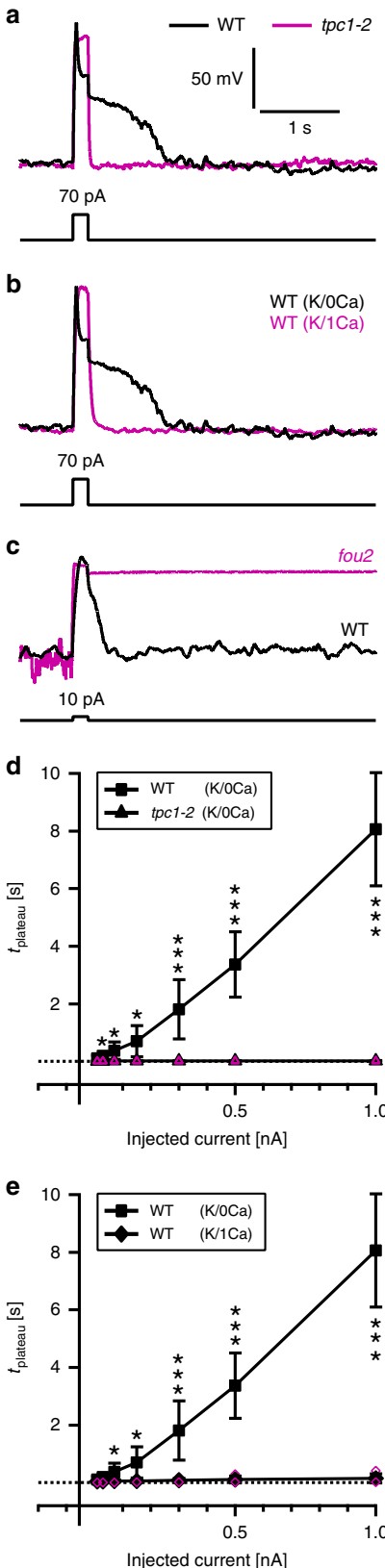

**Fig. 2** Dependence of vacuolar electrical excitability on TPC1/SV channel function. **a** Representative membrane voltage responses (upper panel) of vacuoles from wild type (WT, K/0Ca) and the TPC1 loss-of-function mutant *tpc1-2* (K/0Ca) to 70-pA current injection (lower panel). The voltage response of wild type and *tpc1-2* is shown in black and magenta, respectively. **b** Representative membrane voltage responses (upper panel) of vacuoles from wild type in the absence (WT, K/0Ca, black) and presence of 1 mM luminal $Ca^{2+}$ (WT, K/1Ca, magenta) to 70-pA current injection (lower panel). **c** Representative membrane voltage responses (upper panel) of vacuoles from wild type and *fou2* (K/0Ca) to 10-pA current injection (lower panel). The voltage response of wild type and *fou2* is given in black and magenta, respectively. Note, similar voltage responses were recorded in each of the other two *fou2*-vacuoles tested. **d**, **e** Lifetime of the post-stimulus plateau phase derived for the experimental conditions shown in **a** and **b** were plotted against the corresponding injected current. The dotted lines in **d** and **e** indicate zero. Closed symbols (WT (K/0Ca): squares, *tpc1-2* (K/0Ca): triangles, WT (K/1Ca): diamonds) and error bars in **d** and **e** represent means ± standard deviation from experiments with different vacuoles. Individual data points for *tpc1-2* in **d** and wild type (K/1Ca) in **e** are given by open magenta symbols. The number of experiments was $n = 5$ for *tpc1-2*, $n = 7$ and $n = 5$ for wild type in the absence (WT (K/0Ca)) and presence of luminal $Ca^{2+}$ (WT (K/1Ca)), respectively. Significant differences are displayed by stars according to the determined *p*-values (* = $p < 0.05$, ** = $p < 0.01$, *** = $p < 0.001$; one-way ANOVA followed by Bonferroni's post-hoc comparison test). Current clamp experiments in **a**–**e** were carried out with $K^+$-based solutions in the absence or presence of luminal $Ca^{2+}$ (K/0Ca, K/1Ca, respectively). For the WT (K/0Ca) condition (control), the voltage traces in **a**, **b** and the data points in **d**, **e** are identical to those shown in Fig. 1a, c. Source data are provided as a Source Data file

hyper-excitability of the *fou2* vacuole membrane compared to wild type (Fig. 2c). When excited with only a 10-pA injection pulse, *fou2* vacuoles already exhibited post-stimulus depolarization plateaus that did not repolarize to −60 mV and lasted for the entire post-stimulus recording time of 9.8 s (Fig. 2c). The findings concerning the excitability of wild type vacuoles in relation to *fou2* on the one hand (Fig. 2a, c) and exposure to $Na^+$ or high luminal $Ca^{2+}$ on the other (Fig. 2b, e and Supplementary Fig. 1e, f), are in line with the electrical properties of the TPC1/SV channels[17,23,24]. They indicate that the voltage sensor of TPC1 controls the electrical excitability of the vacuole membrane via influencing channel activity[11,12,21].

**TPK channels are required for TPC1-dependent excitability.** In addition to the $Ca^{2+}$- and voltage-dependent TPC1/SV channel, *Arabidopsis thaliana* mesophyll vacuoles also express the $Ca^{2+}$-dependent TPK1 channel and the structurally related TPK3 isoform[16,27]. While TPC1 channels conduct $K^+$, $Na^+$ and even $Cs^+$ [28–30], TPK1 is highly $K^+$-selective and therefore not permeable for $Na^+$ or $Cs^+$ [15]. To determine whether and how TPKs may support vacuole excitability, wild type plants and TPK1/TPK3 double-loss-of function mutants were compared side by side. Currents of increasing amplitude were injected and the electrical response monitored (Fig. 3). When wild-type vacuoles were excited by 70-pA injection, the post-stimulus depolarization phase was characterized by a lifetime ($t_{plateau}$) of 0.36 s (Fig. 3). Following a 300-pA injection pulse with wild type vacuoles, the lifetime of the post-stimulus depolarization phase increased to 1.81 s (Fig. 3b). In contrast, the *tpk1/tpk3* double mutant responded to each stimulus strength with a significantly much shorter post-stimulus depolarization lifetime (70 pA: $t_{plateau} = 0.09$ s, 300 pA: $t_{plateau} = 0.28$ s; Fig. 3). Thus, the vacuole membrane of the

This is reflected by the activity of *fou2* TPC1 channels at less positive voltages, i.e. closer to the resting membrane voltage, than the wild type TPC1 in the absence of luminal $Ca^{2+}$ (Supplementary Fig. 2[17,26]). Using *fou2* vacuoles under nominal $Ca^{2+}$-free solutes at the luminal side and 1 mM activating $Ca^{2+}$ at the cytosolic side of the vacuole membrane, we found a

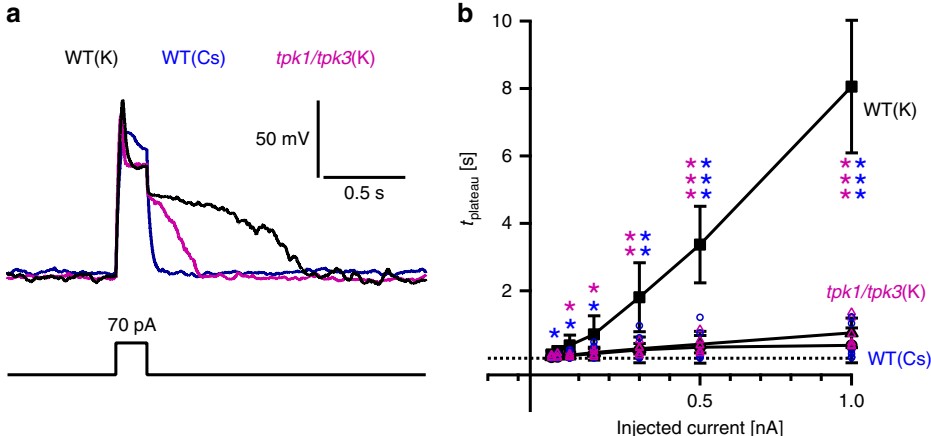

**Fig. 3** Effect of loss of TPK-type channel function on vacuolar electrical excitability. **a** Representative membrane voltage responses (upper panel) of vacuoles from wild type (WT) and the loss-of-function mutant *tpk1/tpk3* to 70-pA current injection (lower panel). Experiments with wild-type vacuoles were conducted either under K$^+$- or Cs$^+$-based solute conditions (WT(K), WT(Cs)) as indicated by black and blue voltage traces, respectively. Voltage responses of *tpk1/tpk3* vacuoles were recorded with K$^+$-based standard solutions and plotted in magenta. **b** Lifetime of the post-stimulus plateau phase derived for the experimental conditions from **a** were plotted against the corresponding injected current. The dotted line in **b** represents the zero line. Closed symbols and error bars in **b** represent means ± standard deviation from experiments with individual vacuoles. Individual data points for WT(Cs) and *tpk1/tpk3*(K) are given by blue and magenta open symbols, respectively. Significant differences between WT(K) vs. WT(Cs) and WT(K) vs. *tpk1/tpk3*(K) are displayed by blue and magenta stars, respectively, according to the determined *p*-values (* = $p < 0.05$, ** = $p < 0.01$, *** = $p < 0.001$; one-way ANOVA followed by Bonferroni's post-hoc comparison test). The number of experiments was $n = 7$ for wild type (K), $n = 7$ for wild type (Cs) and $n = 6$ for *tpk1/tpk3* (K). For the WT(K) condition (control), the voltage trace in **a** and the data points in **b** are identical to those shown in Fig. 1a, c. Source data are provided as a Source Data file

*tpk1/tpk3* mutant was maintained depolarized for a period 4.0 and 6.5 times shorter after the 70 pA and the 300 pA pulse, respectively, than in the wild type background. At current stimulations higher than 300 pA, the difference in the post-stimulus depolarization time between the double mutant and wild type further reinforced. Voltage-clamp current measurements revealed that under K$^+$-based media, voltage-dependent TPC1 channel gating was unaltered and TPC1 currents appeared to be only marginally reduced in the *tpk1/tpk3* double mutant compared to wild type vacuoles (Supplementary Fig. 3a, b). These results indicate that background TPK-mediated K$^+$ fluxes are a prerequisite for TPC1-dependent vacuole excitability. To support this hypothesis, TPKs in wild type vacuoles were challenged with the potassium channel blocker Cs$^+$ (ref. [15]). When K$^+$ was replaced by Cs$^+$ (Supplementary Fig. 3c, d), the post-stimulus depolarization plateau phase was strongly shortened, thus resembling the electrical phenotype of the *tpk1/tpk3* double mutant (Fig. 3). In addition to this knock-out-mimicking effect, Cs$^+$ appeared to have further impact. The lifetime of the post-stimulus depolarization plateau was slightly reduced in Cs$^+$-treated wild type vacuoles compared to *tpk1/tpk3* vacuoles (Fig. 3). This might indicate that besides TPK channels, Cs$^+$ could have affected TPC1 channels as well. Indeed, voltage clamp experiments revealed that the TPC1 conductance was reduced compared to K$^+$-based media (e.g., 3.4-fold at +100 mV), most likely due to the lower permeability of Cs$^+$ over K$^+$ rather than to a change in the voltage-dependent gating behavior (Supplementary Fig. 3a, b)[28]. To further dissect the role of the TPK1 and TPK3 channels in vacuole excitability, the electrical response of the single *tpk1* and *tpk3* mutants was next analyzed. With respect to the lifetime of the post-stimulus depolarization plateau, the *tpk* single mutants did not completely resemble either the wild type or the *tpk1/tpk3* double mutant, but instead showed quite variable intermediate characteristics (Supplementary Fig. 4). This indicates that both channels together contribute to vacuolar excitability. Thus, taken together, our results strongly suggest that

the fraction of the K$^+$ ions that can freely move through open TPK channels affects the TPC1-based excitability.

**Modeling the TPC1/TPK-based membrane excitability**. To gain further insight into the mechanism of vacuole excitability, we modeled vacuole excitability using the cable equation (Supplementary Note 1). In an initial computational cell biology experiment, we included a proton-pump and TPC1, but not TPK, in our model. This basic model allowed us to reproduce the initial two phases of the electrical signal triggered by current injection (Fig. 1a and Supplementary Fig. 5). After the onset of the stimulus, the membrane rapidly depolarized followed by a first partial repolarization. The first repolarization can be explained by the activation of TPC1 at very positive voltages. The outward currents through TPC1 are mainly carried by K$^+$ ions and drive the membrane voltage closer to $E_{K = 0 mV}$, which in turn results in a deactivation of TPC1. After the stimulus, the activity of residual TPC1, however, is not sufficient to maintain the membrane voltage close to $E_K$. In this situation, the proton-pump-driven background conductance dominates and drives the membrane voltage back to the resting level. If the wild type TPC1 channel is replaced by its mutant TPC1-D454N (*fou2*), channel activity around $E_K$ is significantly higher than with the wild type. In the case of *fou2*, the background conductance is outcompeted by the hyperactive TPC1 mutant, clamping the membrane voltage close to $E_K$. Although, the basic model could explain the onset of vacuolar excitation, it failed to reproduce the post-stimulus plateau (Supplementary Fig. 5e). To rectify this, we next included TPK in the model. When we simulated the stimulus-induced depolarization with basal, unregulated TPK activities, we observed that an increasing basal TPK activity reduced the excitability of the membrane but did not reproduce the experimental results (Supplementary Fig. 6a). TPK activity triggered by the external stimulus could not solve this issue either (Supplementary Fig. 6b). This result is in accordance with the experimental finding, that in the absence of TPC1 activity, the depolarizing stimulus did not directly activate the TPKs. Neither

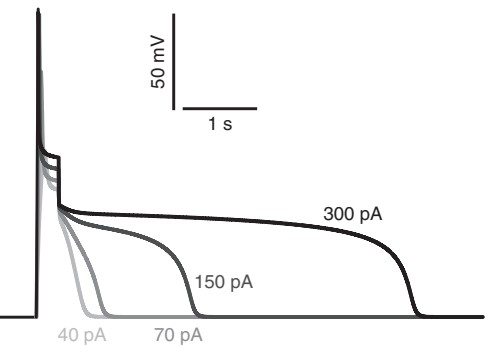

**Fig. 4** Simulation of vacuolar electrical excitability produced by TPC1 and transiently activated TPK. The electrical properties of the vacuole membrane were simulated with background conductance, TPC1 conductance and a TPC1 activity-dependent TPK conductance. TPK channels activate proportionally to the current flowing through TPC1 and inactivate slowly at positive voltages but more rapidly at negative voltages. Current stimuli of 200 ms duration and different intensity were applied to excite the membrane: 40, 70, 150, and 300 pA (cross-referenced with the curves by the respective gray scale). Source data are provided as a Source Data file

wild type vacuoles under TPC1-blocking conditions nor *tpc1-2* vacuoles showed the characteristic TPK-dependent post-stimulus plateau (Fig. 2a, b). Thus, we had to assume a TPC1-dependent TPK-activation upon depolarization. Indeed, when assuming that the TPKs are activated proportionally to the current flowing through TPC1 channels, the experimental data could be reproduced by the model (Fig. 4). To reflect the low basal activity of TPK at resting voltages, we further had to assume that TPKs are inactivated in a voltage-dependent manner, with very low inactivation rates at positive voltages and increasing inactivation rates at more negative voltages. It should be emphasized that a simple dependency of the TPK activity on the voltage amplitude did not allow replication of experimental findings, in particular those obtained with the *tpc1-2* loss-of-function mutant. It was necessary to couple TPK-activation with the activity of TPC1.

## Discussion

Here, we report that in the model plant Arabidopsis the voltage- and $Ca^{2+}$-gated, non-inactivating cation channel TPC1/SV is essential for electrical excitation of the plant vacuole and acts in concert with TPK-type $K^+$-selective channels. An elevated cytosolic $Ca^{2+}$ level will not only open TPC1/SV channels but also TPK channels[6,14] facilitating vacuole excitability. Thus, similar to the tonoplast action potential of excitable cells of charophyte algae[31,32], vacuole excitability of *A. thaliana* mesophyll cells very likely depends on a rise in cytosolic $Ca^{2+}$.

As AtTPC1 is a $Ca^{2+}$-activated $K^+$ channel, all experiments in this work were conducted under elevated 1 mM $Ca^{2+}$ at the cytosolic side of the vacuole membrane. In vivo, however, at a resting $Ca^{2+}$ level of ~0.1 μM, a prominent number of TPC1 channels will not open upon moderate voltage-stimulation[33]. Under this condition, even with a TPK background conductance, the vacuole membrane will rest in the $H^+$ pump-dominated hyperpolarized state. However, a wave of elevated cytosolic $Ca^{2+}$ [34] activates an increasing number of TPKs and a small fraction of TPC1 channels, which weakens but does not yet destabilize the hyperpolarized state. At a membrane resting voltage of around −30 mV[18], the TPC1 channel remained deactivated even when cytoplasmic $Ca^{2+}$ concentrations were elevated up to 1 mM[33]. To destabilize the resting state, both elevated cytosolic $Ca^{2+}$ and a depolarizing electrical stimulus are needed.

In our current clamp measurements, we mimicked the membrane depolarization of the vacuole membrane by current injections. The stronger the stimulus, or in other words, the more the vacuole was depolarized during the current injection period, the more TPC1 channels were activated. Interestingly, the voltage stimulus did not activate TPK channels. Owing to the large density of TPC1/SV channels in the vacuole membrane and the enormous single channel conductance of 80 pS[26], the vacuole membrane will behave like a $K^+$ electrode when reaching or exceeding a critical number of open TPC1/SV channels. The more TPC1/SV channels are open in the post-stimulus period, the more the membrane will be clamped at an $E_K$ of 0 mV. However, since there is a low probability that TPC1/SV channels will be open at a membrane voltage of 0 mV, TPC1 channels will deactivate over time, one after the other. If the number of open, depolarizing ($E_K$ stabilizing) TPC1/SV channels drops below a certain threshold, the hyperpolarizing injected currents, which in vivo are generated by the vacuolar proton pumps, can repolarize the membrane to the pre-stimulus level. Thus, stronger pump-driven repolarization will result in more TPC1/SV channels closing at a faster speed.

The experimentally observed features of the vacuolar electrical excitability could be reproduced in computational cell biological simulations using a model with a hyperpolarizing background conductance, a current stimulus and the two channel types TPC1 and TPK. We had to assume, however, that TPK channels are only transiently active at positive voltages. If we considered a static TPK activity, we just reduced membrane excitability (Supplementary Fig. 6a). In this case, TPKs serve as an additional voltage-stabilizing background conductance. This also holds true if the TPK activity is triggered by the depolarization stimulus and is then maintained active (Supplementary Fig. 6b). Thus, we had to postulate that (i) at resting voltages TPK channels are not active or only marginally active and (ii) the assumed largely voltage-independent TPK channels are regulated by an as yet unknown factor. This factor appears to depend on the activity of TPC1, because in the *tpc1-2* knockout plants we did not observe the stimulus-induced TPK activation. We could reproduce the triggered electrical excitability well if the unknown factor activated TPK proportionally to the stimulus-induced TPC1 activity and if the TPK activity decayed in a voltage-dependent manner with increasing inactivation rate at more negative voltages (Supplementary Note 1). Thus, TPK channels have to be considered as a security valve, which is transiently active at very positive voltages, only. Until recently, it was unimaginable to assign a dynamic component to TPK channels. However, the group of Frans Maathuis has nicely demonstrated that a receptor-like protein kinase activates TPK1 by phosphorylation[35]. Furthermore, salt stress was reported to result in phosphorylation of TPK1 via a $Ca^{2+}$-dependent protein kinase[36]. Thus, one may speculate that in our system the TPK1-regulating protein kinase is modulated in a TPC1-dependent manner.

The voltage responses of the *tpk1/tpk3* double mutant clearly points to a role of TPK-type channels in vacuole excitability. Interestingly, the *tpk1* and *tpk3* single mutants showed highly variable voltage responses, which were not always intermediate between those of the wild type and the double mutant (Supplementary Fig. 4). The stimulus spike amplitude of the *tpk1* mutant, for instance, was significantly smaller than those of wild type, *tpk3* single, and *tpk1/tpk3* double mutant (Supplementary Fig. 4b). These results indicate that the effects of the double mutant cannot be explained simply by a sum of the effects caused by the missing TPK channel function. Thus, TPK1 and TPK3 are not likely to be two independent channels in the vacuole membrane but seem to affect each other. In the plant, these two

different TPK subunits may form heteromeric TPK channels, as shown for voltage-gated plant K$^+$ channels[37].

Human TPC1 channels operate as voltage-dependent Na$^+$ channels, and confer excitability to mouse endolysosomes only in Na$^+$-based media[38]. In contrast, Arabidopsis mesophyll vacuoles expressing TPC1 can be excited in the presence of K$^+$ only. When the luminal side of the AtTPC1 channel protein faced elevated Na$^+$ levels, TPC1/SV channel activity and, in turn, vacuole excitability, were suppressed. Furthermore, here we could demonstrate that even at stimulatory high cytosolic Ca$^{2+}$ conditions (1 mM), a luminal Ca$^{2+}$ concentration of as low as 1 mM completely silenced the TPC1/SV wild-type channel on a scale that was sufficient to effectively suppress vacuole excitability. This indicates that the Ca$^{2+}$ level in the vacuole lumen is important. According to measurements with vacuoles of *Eremosphaera viridis*, *Riccia fluitans*, *Zea mays* roots and *Beta vulgaris* taproots, the physiological luminal free Ca$^{2+}$ concentration ranges from 0.2 to 2.3 mM[33,39,40]. Therefore, to prevent luminal Ca$^{2+}$ from reaching critical levels for TPC1-dependent excitation, tight control of Ca$^{2+}$ loading by CAX-type transporters has to be postulated. In summary, changes in cytosolic and luminal Ca$^{2+}$ concentration, together with the H$^+$ pump-dependent membrane potential, tune TPC1- and TPK-dependent vacuole excitability.

An interesting finding is the striking correlation between input strength and TPC1/TPK-dependent vacuole excitation (Fig. 1c). This relationship might suggest a physiological implication of the vacuole membrane excitability. The response of the vacuole is to a large extent proportional to the incoming stimulus, which is an essential condition for amplifying signals without changing their information content. Different parts of a plant communicate with each other via electrical signals propagating at the plasma membrane that are accompanied by calcium- (cytosol), ROS (extracellular) and potentially also potassium (extracellular) waves[1,34,41–43]. Considering the cable effects and diffusion resistance within the plant body, electrical and chemical signals without amplification on the run would not reach far but would instead fade quickly[1]. A vacuolar mechanism that reacts proportional to a signal input would thus perfectly suit the needs for reamplification.

We still do not know how the propagating electrochemical signals are coupled to the tonoplast. A direct electrical coupling of both membranes via physical contact is rather unlikely, because the plasma membrane is electrically isolated from the vacuole membrane[18]. However, the plasma membrane together with the tonoplast form a sandwich of two membranes with the cytosol in between. Systems of such membrane aggregates are known for surprising dynamic coupling features[44–47]. Electrical changes at one membrane provoke transmembrane ion transport and thus concentration changes in the intermembrane space, which then can induce electrical changes at the other membrane. The molecular understanding of the mechanistic interplay between chemical and electrical signals for plasma membrane-vacuole communication[48] represents a major future challenge.

## Methods

**Plant material**. *Arabidopsis thaliana* ecotype Columbia (Col0) and channel-loss-of-function mutants thereof (*tpc1-2*, *tpk1/tpk3* double mutant, *tpk1-1*, and *tpk3-2* single mutants) were cultivated for 4 to 5 weeks in a growth chamber under controlled conditions before mesophyll protoplasts were enzymatically isolated. For this, fully developed rosette leaves with detached lower epidermis were incubated for 45–60 min at room temperature in an enzyme solution consisting of 0.5% (w/v) cellulase Onozuka R-10 (Serva, Heidelberg, Germany), 0.05% (w/v) pectolyase (Seishin Corp., Tokyo, Japan), 0.5% (w/v) macerozyme R10 (Serva, Heidelberg, Germany), 1% (w/v) bovine serum albumin (Sigma-Aldrich), 1 mM CaCl$_2$, 10 mM Hepes/Tris (pH 7.4) and adjusted to an osmolality of 400 mosmol kg$^{-1}$ with sorbitol. The suspension was filtered through a 50-μm nylon mesh and washed with 400 mM sorbitol and 1 mM CaCl$_2$. After centrifugation at 4 °C and 60 × *g* for 7 min, the precipitated protoplasts were stored on ice until vacuoles were released

under hypo-osmotic solute conditions (10 mM EGTA, 10 mM Hepes/Tris pH 7.4, 200 mosmol kg$^{-1}$ with sorbitol) and used for patch clamp measurements.

The *tpk1/tpk3* double mutant was generated by crossing the corresponding T-DNA insertion lines *tpk1-1* (SALK_146903)[15] and *tpk3-2* (SALKseq_61131). The genotypes of single and double *tpk* mutants were characterized by PCR with the following primers: *TPK1-fwd* (5′-AAA TGT CGA GTG ATG CAG CTC-3′), *TPK1-rev* (5′-TCA AGT TGC TCG AAC TCA TCC-3′), *TPK3-fwd* (5′-ACG TTT CAC GTT CCT CCT CT-3′), *TPK3-rev* (5′-GTT TTG GAT CGG TGA AGA GC-3′), *LBb1.3-rev* (5′-ATT TTG CCG ATT TCGGAAC-3′).

The SALKseq_61131 insertion was recovered from the parental line SALK_61131 using an adapter-ligation PCR approach[49] modified for next-generation sequencing[50]. The primary modification was fragmentation using hydrodynamic shearing of the DNA (Covaris M220) instead of restriction digestion. This step was followed by DNA end-repair, addition of an adenosine overhang, and ligation of a custom universal DNA adapter using the TruSeq kit (Illumina, San Diego, CA). All reactions were performed according to the manufacturer's instructions (TruSeq kit) with the exception that the universal DNA adapter was substituted for the Illumina adapter sequences. PCR amplification (25 cycles) was performed with Phusion polymerase (NEB, Ipswich, MA) and primers specific to the universal DNA adapter and the T-DNA left border sequence. PCR primers contained Illumina compatible adapter sequences, including flowcell attachment sequence, sequencing primer sites, and index sequences. All adapter and primer sequences are provided in Table SI of Hsai et al.[50] and were purchased from IDT (IDT, Coralville, IA). As the parental line had no assigned T-DNA insertion, the newly identified SALKseq_61131 is the only characterized T-DNA insertion in this line. Accession numbers of the studied channels are the following: AtTPC1 (At4g03560), AtTPK1 (At5g55630), AtTPK3 (At4g18160).

**Patch clamp experiments**. All patch clamp experiments were carried out in the whole-vacuole configuration[17,26]. Using an EPC-800 and EPC-10 amplifier (HEKA, Lambrecht, Germany), voltage clamp and current clamp measurements were performed according to the convention for electrical measurements on endomembranes[51]. Therefore, tonoplast membrane voltages referred to the electrical potential at the cytosolic side of the membrane with the luminal potential set to zero. The patch clamp technique was applied to vacuoles of similar size, which was estimated by the compensated membrane capacitance ($C_m$) in the voltage clamp mode being in the range of 8 pF to 35 pF. In the voltage clamp mode, macroscopic currents were recorded by instantaneously applying voltage pulses of 1-s duration, in the range of −80 to +110 mV in 10-mV steps. The holding voltage was −60 mV. Current-density/voltage curves ($I_{ss}/C_m(V)$) were determined by dividing the steady-state current $I_{ss}$ measured at the end of the voltage pulse through the compensated membrane capacitance[17]. Conductance/voltage curves ($G/G_{max}(V)$) were derived from tail current amplitudes and fitted with a Boltzmann Equation (1) representing the gating-scheme described in the Supplementary Note 1 with fixed apparent gating charges ($z_1 = 1.26$; $z_2 = 1.00$).

$$G(V) = \left( \frac{1}{1 + e^{-z_1 \frac{F}{RT}(V - V_1)}} \right)^2 \cdot \left( \frac{1}{1 + e^{-z_2 \frac{F}{RT}(V - V_2)}} \right)^2 \tag{1}$$

In current clamp experiments the vacuole membrane was first set to −60 mV by clamping the current to an appropriate level. In the following, instantaneous current injection pulses ranging from 10 up to 1000 pA were applied for 200 ms before returning to the initial current clamp state. In the voltage and current clamp mode data points were acquired with a sampling rate of 10–200 μs at a low-pass filter frequency of 2.9 or 3 kHz. Data acquisition and off-line analysis were performed with the software programs Pulse, Patchmaster (HEKA Electronik, Lambrecht, Germany), IgorPro (Wave Metrics Inc., Lake Oswego, OR, USA) and Origin2018 (OriginLab Corp., Northampton, MA, USA). The stimulus-spike and the stimulus-plateau voltage amplitude was determined at the beginning and end of the voltage response to the current injection pulse, respectively. The lifetime of the post-stimulus depolarization phase ($t_{plateau}$) indicates the time at which 50% of the initial depolarized voltage level repolarized to the holding voltage (Fig. 1a). The patch clamp data were statistically analyzed with one-way ANOVA followed by Bonferroni's post-hoc comparison test (Supplementary Data 1 and 2). Compared datasets with *p*-values < 0.05 were considered to be significantly different.

If not otherwise stated, the standard bath solution (cytosol) was composed of 150 mM KCl, 2 mM MgCl$_2$, 10 mM Hepes (pH 7.5/Tris) and adjusted to 400 mosmol kg$^{-1}$ with D-sorbitol. The standard pipette solution (vacuolar lumen) was the same with the addition of 0.1 mM EGTA. To specifically examine the role of the TPC1 and TPK channels in vacuole excitability, the Ca$^{2+}$-induced suppression of the fast vacuolar K$^+$-permeable channels (FV[14]) and the Ca$^{2+}$-dependent activation of the TPC1 and TPK channels was ensured by adding 1 mM CaCl$_2$ to the standard bath medium. For some experiments (i) KCl in standard bath and pipette solutions was replaced by equal amounts of either CsCl or NaCl, or (ii) the bath medium contained 1 mM CaCl$_2$ instead of 0.1 mM EGTA.

**Reporting summary**. Further information on research design is available in the Nature Research Reporting Summary linked to this article.

## Data availability

The datasets generated during the current study are available from the corresponding authors on reasonable request (electrophysiological data: Rainer Hedrich; computational simulation data: Ingo Dreyer). The source data underlying Figs. 1–4 and Supplementary Figs. 1–7 are provided as a Source Data file. A reporting summary is available as a Supplementary Information file.

## Code availability

The mathematical algorithms used for computational simulations are described in detail in the Supplementary Note 1.

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

## Acknowledgements

The work was supported by a grant from the European Research Council under the European Union's Seventh Framework Program to R.H. (EU250194), a grant from the German Research Foundation (DFG) within the research group FOR1061 to R.H., a grant from the Chilean Fondo Nacional de Desarrollo Científico y Tecnológico (http://www.conicyt.cl/fondecyt) to I.D. (No. 1150054) and a doctoral fellowship from the China Scholarship Council to J.L. This publication was also funded by the DFG and the University of Würzburg in the funding programme Open Access Publishing.

## Author contributions

R.H. and I.M. designed research. D.J., J.L. performed the electrophysiological experiments, and D.J. analyzed data. I.D. conducted the in silico experiments. R.O. and J.D. generated the *tpk3* and *tpk1/tpk3* loss-of-function mutants, respectively. R.H., I.M., and I.D. wrote the manuscript.

## Additional information

**Competing interests:** The authors declare no competing interests.

