## [Peer Review File · Nature Communications]

Reviewers' comments:

Reviewer #1 (Remarks to the Author):

The manuscript "Voltage-dependent gating of SV channel TPC1 confers vacuole excitability" describes that mesophyll vacuole of the plant cell senses and controls the membrane potential essentially via SV (TPC1) and VK (TPK) channels. The central vacuole occupies about 95% of the cell volume in plants, plays vital roles in cell signaling and cell volume regulation, and facilitates various metabolisms among others. The mechanisms underlying the vacuolar membrane excitability and the molecular components involved remain largely unknown. In this manuscript, Jašlan et al. applied the patch clamp technique to test electrical excitability of the central vacuole, and concluded that the vacuolar K⁺-conducting TPC1 and TPK channels act in concert to mediate Ca²⁺- and voltage-induced electrical excitability to the central organelle of plant cells. This finding is novel and important, as new functions are assigned to these two channels, for which their functions are unclear largely. Overall, I think that this manuscript describes important breakthroughs and contains interesting data that are very convincing. With respect to the molecular mechanisms, this work is significant since it suggests that TPK1-type channels is required for TPC1-dependent excitability. It is also important that the authors propose the novel physiological relevance for the TPC1/TPK-based membrane excitability, such as response to Na⁺ stress.

Major comments:

1. The authors used TPK1/TPK3 double-loss-of function mutant in this work because TPK1 and TPK3 are both expressed in mesophyll vacuoles. How about the *tpk1* or *tpk3* single mutant? Are they redundant or not?
2. This work was conducted under elevated 1 mM Ca²⁺ at the cytosolic side of the vacuole membrane. As stated by the authors, 1 mM Ca²⁺ is extremely high for cytosolic side, a range of cytosolic Ca²⁺ concentrations in conjunction with a range of cytosolic K⁺ concentrations should be analyzed to validate/establish the function of SV in the excitability under conditions close to physiological levels of Ca²⁺ and K⁺.
3. The numbers of vacuole patched were 3 to 7, which seem lower for electrophysiological experiments, considering the large variations. In addition, no statistical P values are given in the figure legends or the main text, which makes the reading difficult. Clearly, some experiments were re-analyzed throughout the figures, including traces.

Minor comments:

4. In the introduction, the TPK family should be introduced, i.e. how many homologs/isoforms exist in the genome, what the biophysical properties of these TPKs are.
5. Supplementary Fig.6 figure legends b should be c.
6. The sentence from line 122 to 124 should be modified, and the refer figure or reference should be added.

Reviewer #2 (Remarks to the Author):

The paper by Jaslan et al analyzes the excitability of the vacuolar membrane in mesophyll cells. It specifically focuses on the role of TPC1, a non-selective cation channel that constitutes the main conductance of the tonoplast. By using a combination of current and voltage clamp, the authors convincingly show that isolated vacuoles are 'excitable' and that TPC1 is instrumental in this process though a second cation conductance (TPK1) is also involved.

This is an interesting paper with a novel approach to study vacuolar 'action potentials' in plant cells. However, since tonoplast excitability has been studied previously in giant algae (e.g. by Kikuyama and colleagues, Findlay and colleagues, and so forth) one could argue that the novelty

mainly derives from using an alternate system rather than describing a 'new' phenomenon.

Recordings and modeling were carried out with rigor and the presented core findings visualize some important aspects of the tonoplast 'action potential' but at the same time the reader is left with many questions both at the technical and conceptual level.

Technical:

- (i) Can the authors give any justification for the magnitude of the used current injections, especially in relation to estimates for pumps rates.
- (ii) Should the fou response not be same as TPC1 WT with 0 luminal Ca (figure 2)?
- (iii) How come the same current gives such comparable depolarizations, e.g. figure 2a for WT and tpc1. Surely, the membrane conductance of tpc1 vacuoles is way lower than that of WT vacuoles and thus tpc1 vacuoles should be depolarized deeper.

Conceptual:

- (i) What is the physiological role of tonoplast excitability?
- (ii) Secondly, instead of the artificial use of direct current injection, what would be the physiological upstream trigger? Is Ca the only option?
- (iii) Is there evidence in intact mesophyll cells for excitability and if so how is polarization of the plasma membrane coupled to the tonoplast? Kikuyama and others proposed Ca as the plasma membrane-tonoplast coupling trigger but what about alternates such as electrical coupling?
- (iv) The authors propose on page 9 that Ca is the most likely coupling/trigger device but do not provide any evidence in spite of a wealth of data in the literature on TPC1 and TPK1 Ca dependence.
- (v) Another question concerns the (potential) role of anion conductances. Several authors propose that an early (Ca-induced) Cl current initializes the tonoplast depolarization (e.g. Kourie 1994). Again a similar process may occur in higher plants.
- (vi) Line 188: 'To achieve such a dependency, TPKs were modelled to be activated proportionally to the current flowing through TPC1 channels.' Is there any experimental evidence to support such a scenario?
- (vii) Line 247: 'This means, contrary to the previous view, TPK1 channel activity should alter with voltage, as reported for the TPK4 channel'. See above.

Point-to-point responses to the comments of the referees to the manuscript entitled “Voltage-dependent gating of SV channel TPC1 confers vacuole excitability” (NCOMMS-18-36636)

Response to the comments of Reviewer #1

1) **Rev#1.1:** The authors used TPK1/TPK3 double-loss-of function mutant in this work because TPK1 and TPK3 are both expressed in mesophyll vacuoles. How about the *tpk1* or *tpk3* single mutant? Are they redundant or not?

Answer: To answer this open and very interesting question, we performed further patch clamp measurements with single *tpk1* and *tpk3* mutants. With respect to the lifetime of the post-stimulus depolarization plateau, the *tpk* single mutants did not completely resemble either the wild type or the *tpk1/tpk3* double mutant, but instead showed quite variable intermediate characteristics (new Supplementary Fig. 4a). This indicates that both channels together contribute to vacuolar excitability. Interestingly, the *tpk1* and *tpk3* single mutants showed highly variable voltage responses, which were not always intermediate between those of the wild type and the double mutant (new Supplementary Fig. 4). The stimulus spike amplitude of the *tpk1* mutant, for instance, was significantly smaller than those of wild type, *tpk3* single and *tpk1/tpk3* double mutant (new Supplementary Fig. 4b). These results indicate that the effects of the double mutant cannot be explained simply by a sum of the effects caused by the missing TPK channel function. Thus, TPK1 and TPK3 are not likely to be two independent channels in the vacuole membrane but seem to affect each other. In the plant, these two different TPK subunits may form heteromeric TPK channels, as shown for voltage-gated plant K⁺ channels (Lebaudy et al. 2008; Plant J; doi: 10.1111/j.1365-313X.2008.03479.x.).

These new data sets are now described and discussed in the revised manuscript.

2) **Rev#1.2:** This work was conducted under elevated 1 mM Ca²⁺ at the cytosolic side of the vacuole membrane. As stated by the authors, 1 mM Ca²⁺ is extremely high for cytosolic side, a range of cytosolic Ca²⁺ concentrations in conjunction with a range of cytosolic K⁺ concentrations should be analyzed to validate/establish the function of SV in the excitability under conditions close to physiological levels of Ca²⁺ and K⁺.

Answer: In the past, the functional properties of the SV/TPC1 channel have been studied in detail in the presence of various Ca²⁺ and K⁺ concentrations and gradients (e.g. Ivashikina and Hedrich 2005, doi.org/10.1111/j.1365-313X.2004.02324.x; Dadacz-Narloch et al. 2011, doi.org/10.1105/tpc.111.086751; Potossin et al. 2005, doi.org/10.1007/s00232-005-0766-3, Carpaneto and Gradogna 2018, doi.org/10.1016/j.bpc.2018.02.006, Larisch et al. 2016, doi.org/10.1007/s00018-016-2131-3). Since this vacuolar channel cannot be patched inside the *A. thaliana* mesophyll cell, all labs have studied TPC1 in isolated vacuoles and mimicked

physiological conditions using symmetric 100 mM K⁺. In the presence of resting nanomolar cytosolic Ca²⁺ levels, TPC1 is closed. Under stress, the cytosolic Ca²⁺ level rises several-fold. Because of limited resolution (a technical problem) no one has yet measured the concentration of the stress-peak Ca²⁺ level at the cytosolic side of the vacuolar membrane in planta. Therefore, to find out the role of active SV/TPC1 channels, all researchers have previously used 0.1-1 mM Ca²⁺ in the cytosolic buffer. In the present work, the experiments were carried out in the presence of 1 mM cytosolic Ca²⁺ to ensure the inhibition of the FV (fast vacuolar) channels (Allen and Sanders, 1996, doi.org/10.1046/j.1365-313X.1996.10061055.x) and the activation of the TPC1 and TPK channels, to specifically examine the role of these channel types in vacuole excitability. Under these cytosolic Ca²⁺ conditions, we could show that whenever TPC1 activity is strongly reduced by a rise in luminal Ca²⁺ or replacement of K⁺ by Na⁺, vacuole excitability is suppressed.

The reasons for using 1 mM cytosolic Ca²⁺ (bath medium) are now given in the Material and Method section of the revised manuscript as follows:

“To specifically examine the role of the TPC1 and TPK channels in vacuole excitability, the Ca²⁺-induced suppression of the fast vacuolar K⁺-permeable channels (FV¹⁴) and the Ca²⁺-dependent activation of the TPC1 and TPK channels was ensured by adding 1 mM CaCl₂ to the standard bath medium.”

3) Rev#1.3: The numbers of vacuole patched were 3 to 7, which seem lower for electrophysiological experiments, considering the large variations. In addition, no statistical P values are given in the figure legends or the main text, which makes the reading difficult. Clearly, some experiments were re-analyzed throughout the figures, including traces.

Answer: Considering the observed variations of the data, the analysis of 3-7 vacuoles for each condition allowed already unequivocal conclusions as indicated by the P values. Nevertheless, we carried out additional current and voltage clamp experiments (e.g. for wild type, *tpc1-2* and *tpk1/tpk3*, the number of current clamp experiments under K-based solute conditions were increased from n=3 to n= 7 (WT), n=5 (*tpc1-2*) and n=6 (*tpk1/tpk3*)), and these results were still consistent with our previous ones. To make the reading easier, the significance of *p* values is now provided in the figures or figure legends. For clearer presentation, the raw traces and analyzed data points for different plant lines or solute conditions were compared with WT(0/Ca) in separate graphs instead of superimposing them in one graph. In the revised manuscript, we now mentioned that the identical raw traces/data points from WT(0/Ca) from Fig. 1 were used for these comparisons in the other figures.

4) Rev#1.4: In the introduction, the TPK family should be introduced, i.e. how many homologs/isoforms exist in the genome, what the biophysical properties of these TPKs are.

Answer: The referee is right. In the revised manuscript we inform the reader about the TPK channel family already in the introduction as suggested:

“In addition to TPC1 channels, tandem-pore potassium (TPK) channels are also equipped with EF hand motifs. They are activated upon a rise in the cytosolic Ca^{2+} concentration^{14,15}, and in part contribute to the vacuole Ca^{2+} -dependent K^{+} conductance^{15,16}. Among the four vacuolar-localized TPK family members, TPK1 and TPK3 are expressed in mesophyll cells (*Arabidopsis* eFP browser, <http://bar.utoronto.ca/efp/cgi-bin/efpWeb.cgi>).”

5) Rev#1.5: Supplementary Fig.6 figure legends b should be c.

Answer: This typo has been corrected.

6) Rev.#1.6: The sentence from line 122 to 124 should be modified, and the refer figure or reference should be added.

Answer: In the revised manuscript the requested information is provided:

“The findings concerning the excitability of wild type vacuoles in relation to *fou2* on the one hand (Fig. 2a, c) and exposure to Na^{+} or high luminal Ca^{2+} on the other (Fig. 2b, e, Supplementary Fig. 1e, f), are in line with the electrical properties of the TPC1/SV channels^{17,23,24}.”

Response to comments of reviewer #2:

1) Rev#2.1: “This is an interesting paper with a novel approach to study vacuolar ‘action potentials’ in plant cells. However, since tonoplast excitability has been studied previously in giant algae (e.g. by Kikuyama and colleagues, Findlay and colleagues, and so forth) one could argue that the novelty mainly derives from using an alternate system rather than describing a ‘new’ phenomenon.”

Answer: We agree with the referee stating that plasma membrane and vacuole action potentials (APs) have been discovered in giant algae first and have been investigated in this organism in great detail. In the revision, we now underline this fact better. By contrast to Chara, Chara-type all-or-nothing cell membrane APs are not fired in higher plants like *Arabidopsis thaliana* (At), and a vacuole excitability has not been known until now. Thus, the outcome of our study – that At vacuoles can be excited – was not expected and therefore is novel “describing important breakthroughs” as stated by Rev#1. Furthermore, we did not just report *Arabidopsis* vacuole excitability but identified two molecular key players that are essential for vacuole excitability in the model plant for the first time. Such a breakthrough at the molecular level is still awaited for the Chara system. Our findings revealed that different ion channel entities underly vacuole excitability, such as potassium-permeable channels (TPC1, TPKs) in *Arabidopsis* rather than anion channels in Chara (for more details see below and answer to comment 2.9). Just very recently, genome and transcriptome sequence information of Chara became available (Nishiyama et al., 2018, doi.org/10.1016/j.cell.2018.06.033, Cahoon et al., 2017, doi.org/10.3390/genes8020080). Based on these new resources, transformation/gene KO protocols can now be established for Chara

and the role of TPC1 and TPKs can be studied in giant algae in the near future. In the revision, we now provide more information about the current situation in Chara as follows:

In **Introduction** (page 3 and 4):

“Excitability of the vacuole membrane was, however, recognized in Charophyceae freshwater green algae. In Chara, the tonoplast action potential is coupled to the excitation of the plasma membrane via a cytosolic Ca^{2+} signal^{3,4}, which stimulates the release of Cl^- ions from the vacuole and in turn leads to a transient vacuole polarization. The molecular nature of the ion channel activities underlying the Chara-type action potentials at the plasma membrane and vacuole membrane have remained unknown.”

2) Rev#2.2_technical_(i) Can the authors give any justification for the magnitude of the used current injections, especially in relation to estimates for pumps rates.

Answer: Proton pumping rates in *Arabidopsis thaliana* mesophyll vacuoles determined by Rienmüller et al. (2012, doi.org/10.1074/jbc.M111.310367) are in the range of 1-3 pA/pF. An average size vacuole has a capacitance of about 20 pF. Thus, a current injection of 20-60 pA is necessary to compensate for the endogenous pump currents. This is well in line with the experimental observation that stimuli of about 70 pA (which corresponds to 3.5 pA/pF) can induce vacuole excitability (Fig. 1a, c). The maximum stimulus applied was 1000 pA which corresponds to 50 pA/pF (Fig. 1c). This value is also far below the maximal currents measured for TPC1 under voltage clamp conditions (about 500 pA/pF) (Supplementary Fig. S2a, b). Thus, the magnitude of the current injections imposed is well within the physiological range.

The readers will find this information in the revised manuscript.

3) Rev#2.3_technical_(ii) Should the fou response not be same as TPC1 WT with 0 luminal Ca (figure 2)?

Answer: No. The impaired luminal Ca^{2+} binding site of *fou2* not only leads to an insensitivity of the TPC1 channel to luminal Ca^{2+} but, at the same time, to a fundamental change in voltage-dependent channel gating. As a result, voltage-dependent activation of *fou2* TPC1 channels is shifted to more negative membrane voltages. This means that less depolarized membrane voltages are required to activate the *fou2* TPC1 channels. Therefore, in contrast to wild type TPC1 channels, a weak injection pulse (10 pA) is sufficient to activate a critical number of *fou2* TPC1 channels that the depolarization phase is maintained for the entire post-stimulus recording time of 9.8 s. To make this aspect clearer for the reader, we revised the related sentence on page 5/line 115-117 as followed:

“The *fou2* point mutation, D454N in TPC1, impairs the luminal Ca^{2+} coordination site^{17,23,25}. As a result, the TPC1/SV channel becomes **not only** insensitive towards inhibitory luminal Ca^{2+} , **but** shows a fundamental change in its voltage-dependent gating features. This is reflected by the

activity of *fou2* TPC1 channels at less positive voltages, i.e. closer to the resting membrane voltage, than the wild type TPC1 in the absence of luminal Ca^{2+} (Supplementary Fig. 2^{17,26}).”

4) Rev#2.4_technical_(iii) How come the same current gives such comparable depolarizations, e.g. figure 2a for WT and *tpc1*. Surely, the membrane conductance of *tpc1* vacuoles is way lower than that of WT vacuoles and thus *tpc1* vacuoles should be depolarized deeper.

Answer: The reviewer is correct when stating that *tpc1-2* vacuoles should be depolarized deeper than WT vacuoles, because the membrane conductance of *tpc1-2* vacuoles is lower than that of the WT. And indeed, this is shown in Fig. 2a. Initially, at the resting membrane voltage, channels of the TPC1 type are closed and the membrane conductance of *tpc1-2* and WT vacuoles is similar. Therefore, the initial stimulus depolarizes the membranes of both to a similar degree; *tpc1-2* vacuoles stay depolarized at this initial level until the end of the stimulus. In contrast, in WT vacuoles, channels of the TPC1-type activate with some delay and provoke a reduced depolarization. At the end of the stimulus, the difference between WT and *tpc1-2* vacuoles is clearly visible (Fig. 2a). A similar difference is visible when blocking TPCs with 1 mM luminal Ca^{2+} (Fig. 2b).

We now explain these points better in the revised manuscript (at the top at page 6).

5) Rev#2.5_Conceptual_(i) What is the physiological role of tonoplast excitability?

Answer: Recent works by Choi et al. (2014, doi.org/10.1073/pnas.1319955111) and Gilroy et al. (2014, doi.org/10.1016/j.tplants.2014.06.013) have shown that the loss or overexpression of TPC1 feeds back on the magnitude and velocity of stress-induced Ca^{2+} waves. As suggested in our TIPS review (Hedrich et al., 2016, doi.org/10.1016/j.tplants.2016.01.016), vacuole excitability will support the release of Ca^{2+} and other compounds from the vacuolar store and will thus contribute to the systemic plant response.

To address this point, we add a new paragraph at the end of the discussion as follows:

“Duration of vacuole excitation depends on stimulus intensity

An interesting finding is the striking correlation between input strength and TPC1/TPK-dependent vacuole excitation (Fig. 1c). This relationship might suggest a physiological implication of the vacuole membrane excitability. The response of the vacuole is to a large extent proportional to the incoming stimulus, which is an essential condition for amplifying signals without changing their information content. Different parts of a plant communicate with each other via electrical signals propagating at the plasma membrane that are accompanied by calcium- (cytosol), ROS (extracellular) and potentially also potassium (extracellular) waves^{1,34,41-43}. Considering the cable effects and diffusion resistance within the plant body, electrical and chemical signals without amplification on the run would not reach far but would instead fade quickly¹. A vacuolar mechanism that reacts proportional to a signal input would thus perfectly suit the needs for reamplification.”

6) Rev#2.6_Conceptual_(ii) Secondly, instead of the artificial use of direct current injection, what would be the physiological upstream trigger? Is Ca the only option?

Answer: Given that both TPC1 and TPKs have EF hands and require a Ca²⁺ rise for activation, we think that Ca²⁺ is the most likely upstream trigger and thus, we do not want to speculate about options not supported by experimental evidence. Since calcium signals very often are associated with ROS production, we cannot exclude that ROS is involved, too. We shortly addressed this item in the new paragraph at the end of the discussion (see also response to comment of **Rev#2.5**)

7) Rev#2.7_Conceptual_(iii) Is there evidence in intact mesophyll cells for excitability and if so how is polarization of the plasma membrane coupled to the tonoplast? Kikuyama and others proposed Ca as the plasma membrane-tonoplast coupling trigger but what about alternates such as electrical coupling?

Answer: Direct electrical coupling is not possible for physical reasons; the plasma membrane is electrically isolated from the vacuole membrane. However, the plasma membrane together with the tonoplast forms a system of two adjacent membranes with the cytosol in between them. Systems with such membrane arrangements show surprising dynamic coupling features (Dreyer et al., 2017, doi.org/10.1111/nph.14667; Dreyer et al., 2019, doi.org/10.1111/nph.15646; Schott et al., 2016, doi.org/10.3389/fpls.2016.00912; Gajdanowicz et al., 2011, doi.org/10.1073/pnas.1009777108). Electric changes at one membrane provoke transmembrane ion transport and thus concentration changes in the intermembrane space, which then can induce electric changes at the other membrane. However, the analysis of these coupling processes would go far beyond the scope of the present study, but will be certainly addressed in the future.

We shortly address this point in the new last paragraph at the end of the discussion as follows:

“We still do not know how the propagating electrochemical signals are coupled to the tonoplast. A direct electrical coupling of both membranes via physical contact is rather unlikely, because the plasma membrane is electrically isolated from the vacuole membrane¹⁸. However, the plasma membrane together with the tonoplast form a sandwich of two membranes with the cytosol in between. Systems of such membrane aggregates are known for surprising dynamic coupling features⁴⁴⁻⁴⁷. Electrical changes at one membrane provoke transmembrane ion transport and thus concentration changes in the intermembrane space, which then can induce electrical changes at the other membrane. The molecular understanding of the mechanistic interplay between chemical and electrical signals for plasma-membrane-vacuole communication⁴⁸ represents a major future challenge.”

8) Rev#2.8_Conceptual_(iv) The authors propose on page 9 that Ca is the most likely coupling/trigger device but do not provide any evidence in spite of a wealth of data in the literature on TPC1 and TPK1 Ca dependence.

Answer: In the past, the functional properties of SV/TPC1 channel have been studied in detail in the presence of various Ca^{2+} and K^+ concentrations and gradients (e.g. Ivashikina and Hedrich 2005, doi.org/10.1111/j.1365-313X.2004.02324.x; Dadacz-Narloch et al. 2011, doi.org/10.1105/tpc.111.086751; Potossin et al. 2005, doi.org/10.1007/s00232-005-0766-3, Carpaneto and Gradogna 2018; doi.org/10.1016/j.bpc.2018.02.006,; Larisch et al. 2016, doi.org/10.1007/s00018-016-2131-3). Since this vacuolar channel cannot be patched inside the *A. thaliana* mesophyll cell, all labs have studied TPC1 in isolated vacuoles and mimicked physiological conditions using symmetric 100 mM K^+ . Under resting nanomolar Ca^{2+} levels TPC1 is closed. Under stress, the cytosolic Ca^{2+} level rises several-fold but nobody has yet measured the concentration of the stress-peak Ca^{2+} level at the cytosolic side of the vacuolar membrane *in planta*. Therefore, to find out the role of active SV/TPC1 channels, all researchers have previously used 0.1-1 mM Ca^{2+} in the cytosolic buffer. In the present work the experiments were carried out in the presence of 1 mM cytosolic Ca^{2+} to ensure the inhibition of the FV (fast vacuolar) channels (Allen and Sanders 1996, Plant J. 10: 1055) and activation of the TPC1 and TPK channels, to specifically examine the role of these channel types in vacuole excitability. Under these cytosolic Ca^{2+} conditions we could show that whenever TPC1 activity is strongly reduced by a rise in luminal Ca^{2+} or replacement of K^+ by Na^+ , vacuole excitability is suppressed.

9) Rev#2.9_Conceptual_(v) Another question concerns the (potential) role of anion conductances. Several authors propose that an early (Ca-induced) Cl^- current initializes the tonoplast depolarization (e.g. Kourie 1994). Again a similar process may occur in higher plants.

Answer: This is a good point and we thought about it as well. Firstly, transient Cl^- and K^+ currents during the action potential in *Chara inflata* (Kourie 1994) suggest the existence of calcium-activated chloride channel, but so far no molecular evidence has supported this hypothesis. In the *A. thaliana* genome there is a single member of the TMEM family, which in animals encode a directly calcium-activated plasma membrane chloride channel. So far, however, no experimental evidence for AtTMEM function exists that allows us to speculate that in plants AtTMEM is localized to the vacuole membrane and functions in the same manner as the animal plasma membrane-restricted TMEMs. Secondly, one has to realize that a different sign convention for the vacuole membrane voltage was used in the former Chara papers to describe tonoplast voltages. When (i) the same sign convention is applied in those Chara publications as in our manuscript and (ii) the cytosol-directed Cl^- gradients in both systems are considered, a Ca-induced Cl^- current should not only initialize a tonoplast hyperpolarization in Chara but also in mesophyll protoplasts. The fact that we, however, observed a tonoplast depolarization rather than a hyperpolarization actually excludes an initial role the Cl^- currents in mesophyll tonoplast excitability.

In this context see also last part of the answer to question Ref#2.1 of referee 2:

In contrast to Chara, we for the first time identified At vacuole excitability together with the major molecular players of it. Just very recently, genome and transcriptome sequence information of Chara became available (Nishiyama et al., 2018, Doi.org/10.1016/j.cell.2018.06.033, Cahoon et al., 2017, doi.org/10.3390/genes8020080). Based on these new resources, transformation/gene KO

protocols can now be established for Chara and the role of TPC1 and TPKs can be studied in giant algae in the future as well. In the revision, we now provide more information about the current situation in Chara.

10) Rev#2.10_Conceptual_(vi) Line 188: ‘To achieve such a dependency, TPKs were modelled to be activated proportionally to the current flowing through TPC1 channels.’ Is there any experimental evidence to support such a scenario?

Answer: The reviewer addresses an important aspect. The data presented in the manuscript support such a scenario; admittedly in an indirect way, however, but no less significant. The comparison of the experimental with the modeling results indicate that the observed phenomenon can neither be explained with a basal TPK activity nor with a triggered activity due to the depolarization stimulus (Supplementary Fig. 5a, b). A similar important finding was that the depolarizing stimulus apparently did not activate the TPKs directly in the absence of TPC1 activity; neither *tpc1-2* vacuoles nor wild type vacuoles with blocked TPC1 show the characteristic TPK-dependent post-stimulus plateau (Fig. 2a, b, red traces). Thus, we had to assume a TPC1-dependent TPK-activation upon depolarization. And indeed, when assuming that the TPKs are activated proportionally to the current flowing through TPC1 channels, the experimental data could be reproduced in the model in — also for us astonishing — detail. It should be noted that a simple dependency of the TPK activity on the voltage amplitude did not allow replication of experimental findings, in particular those obtained with the *tpc1-2* loss-of-function mutant. It was necessary to couple TPK activation with the activity of TPC1. These conclusions are now better explained in the text.

11) Rev#2.11_Conceptual_(vii) Line 247: ‘This means, contrary to the previous view, TPK1 channel activity should alter with voltage, as reported for the TPK4 channel’. See above.

Answer: As outlined in the answer to point 2.10, the presented combination of experimental and modeling approaches provides indirect evidence for the conclusions drawn. Therefore, our study will certainly encourage future experiments from which a direct answer about the model-derived predictions of TPK’s voltage dependent properties will be gained.

Additional changes:

- Since publication in *Nature Communications* does not allow Supplementary Text, we incorporated the former Supplementary Text into the main body of the manuscript.
- We also provided the requested Source Data.

REVIEWERS' COMMENTS:

Reviewer #1 (Remarks to the Author):

Both referees were impressed by the innovative tools utilized and the interesting molecular insights gained in the manuscript "Voltage-dependent gating of SV channel TPC1 confers vacuole excitability". As mentioned by the other referee, the tonoplast excitability has been studied in giant algae, and the comparison between these in higher plants and the Chara should be carried out (an excellent point). In the current study, genetic mutants were used and the modeling approach was employed, which allow these authors to uncover the previously unknown molecular mechanisms underlying the tonoplast excitability in both lower and higher plants. These authors have carefully addressed this referee's concerns as well as other referee's concerns with additional experiments and clarifications. The central vacuole occupies over 90% of the plant cell volume, while its signaling role, equivalent to the ER in animal cells, has not been well established. The current study that intends to tie plasma membrane-originated calcium signaling to the central vacuole might open new avenues, such as long distance signaling in plants (that is still a myth now). Overall, all the concerns from this referee have been addressed and this referee is satisfied with the revision. The referee looks forward to its publication in Nature Communications.

Reviewer #2 (Remarks to the Author):

I am happy with the authors' responses to my queries and have no further comments other than congratulating the authors on a nice piece of work.